A SNP-based genome-wide association study (GWAS) of seed-yield related traits in Psathyrostachys juncea using wheat as a reference genome

Li Zhen 1
Wang Tian 2
Ren Xiaomin 3
Han Feng 1
Ma Yingmei 4
Yun Lan nmg_yunlan@163.com 1
1 College of Grassland Science, Inner Mongolia Agricultural University , Inner Mongolia , China
2 Bayannur Agriculture and Animal Husbandry , Inner Mongolia , China
3 Inner Mongolia University , Inner Mongolia , China
4 College of Desert Control Science and Engineering, Inner Mongolia Agricultural University , Inner Mongolia , China
Abd El-Moneim Diaa
Electronic publication date: 2025 Jul 22
Publication date: 2025
Volume: 13
Electronic Location ID: e19617
Received 2025 Jan 29; Accepted 2025 May 28
Copyright: ©2025 Li et al.
Copyright year: 2025
Copyright holder: Li et al.
License: This is an open access article distributed under the terms of the Creative Commons Attribution License, which permits unrestricted use, distribution, reproduction and adaptation in any medium and for any purpose provided that it is properly attributed. For attribution, the original author(s), title, publication source (PeerJ) and either DOI or URL of the article must be cited.
License URL: https://creativecommons.org/licenses/by/4.0/

Keywords: SNP, Psathyrostachys juncea, Genome-wide association study, Seed yield trait

Funding: National Natural Science Foundation of China 32371762 Youth Fund Project of the Natural Science Foundation of Inner Mongolia of China 2025QN03023 This study was financially supported by the National Natural Science Foundation of China, grant number 32371762, and the Youth Fund Project of the Natural Science Foundation of Inner Mongolia of China grant number 2025QN03023. The funders had no role in study design, data collection and analysis, decision to publish, or preparation of the manuscript.

==============================
Background

Psathyrostachys juncea is an popular perennial grass for both mowing and grazing when used for pasture establishment in high latitude regions. P. juncea has the advantages of high yield, high quality, good palatability and high nutritional value. It is widely used in artificial grassland construction and ecological restoration in Inner Mongolia and other regions.

Methods

The current study aimed to identify genetic signals associated with seed yield in P. juncea germplasm through a genome-wide association study (GWAS) using wheat as a reference genome. 300 accessions of P. juncea germplasm from different countries were used as materials to assess eight seed yield-related traits in two environments for two consecutive years.

Results

All the trait values varied considerably across genotypes. Across different locations, the coefficient of variation among genotypes was the highest for seed yield per plant, which exceeded 70%. Further correlation analysis of seed yield factors showed that seed yield per plant had a significant positive correlation with reproductive tiller number. The population exhibited five population structures (Q) assessed by using 84,024 single-nucleotide polymorphisms (SNPs). After controlling Q and K (subgroups), GWAS identified 121 SNPs significantly associated with eight traits. Among them, 19 SNPs were detected in multiple environments, and a total of 91 candidate genes were annotated, which involve the synthesis of cell wall polysaccharides and proteins, plant growth and development, photosynthesis, gibberellin regulation, hormone signal transduction, phenylalanine metabolism, and amino acid metabolism processes. The identification of SNP signals and related candidate genes could enrich the existing genomic resources and lay a foundation for the study of molecular breeding, mapping, and cloning of important genes in P. juncea.

Introduction

Psathyrostachys is a small genus of Gramineae Triticeae with only about ten species. Psathyrostachys originated in Eurasia, ranging from the Middle East to Russia, across central Asia to Mongolia and northern China. There are four wild species in China, namely P. huashanica Keng ex P. C. Kuo, P. lanuginosa (Trin) Nevski, P. juncea (Fisch.) Nevski and P. kronenburgii (Hack.) Nevski. P. huashanica is an endemic species in China and its distribution is limited to Huashan Mountain in Shaanxi Province. Other species are mainly distributed in Tianshan Mountain and Altai Mountain of Xinjiang and Gansu Province and Qinghai-Tibet Plateau (Dewey, 1984). Most of Psathyrostachys species are perennial cool- season grasses and well-adapted to cold uplands and semiarid deserts. After being introduced into North America P. juncea is also known as Russian wildrye. It is a cross-pollinated, perennial bunch grass with dense clustered leaves, multiple tillers, and strong roots (Asay, 2008). It is the only grass species with feeding value of Psathyrostachys. Due to its strong cold and drought tolerance, as well as maintains high nutritional value in summer and autumn, it is a valuable grass species for forage production and ecological restoration (Xiong et al., 2020). It is mainly used in arid and semi-arid areas of North America for grassland reseeding and dry pasture establishment, as well as for salinized and alkaline grassland improvement. There are multiple strong resistance genes of P. juncea that have been used for wheat improvement, including resistance to powdery mildew, sharp eyespot and yellow stunt virus (Han et al., 2020; Bai et al., 2020; Hu et al., 2018). Therefore, P. juncea is an excellent perennial grass with feeding, ecological and breeding value.

Compared with crops, most perennial forage grasses have strong vegetative reproduction ability, while their sexual reproduction ability is relatively weak (Garcia et al., 2019; Easson, White & Pickles, 1993). The short blooming period and cross-pollination ability of P. juncea lead to erratic and relatively low seed yields (Foulkes et al., 2007). The breeders have to focus on the development of cultivars with higher seed yield and yield stability based on the reproductive characteristics of perennial cross-pollinated plants. Seed yield is controlled by numerous genes that interact with each other and with the environment. Seed yield is a complex trait that is determined by spike number, seed weight per spike, spike length, spike node number and thousand kernel weight, and each component trait is a quantitative trait controlled or affected by multiple loci (Shokat et al., 2020; Gao et al., 2017). These traits have a great influence on improving plant seed yield. Thus, there needs to be a detailed genetic dissection of the seed yield trait of cross-pollinated perennial plants and its component traits to manipulate the alleles at the relevant loci to the greatest advantage.

With the development of high-throughput sequencing technology, more and more plant genome sequences have been published, which is conducive to obtaining high density molecular markers covering the whole genome, facilitating genome-wide association analysis, and enabling the discovery of many agronomic traits related loci. As a critical breeding focus, the research on molecular mechanism of seed yield traits has application value in varia-type of plants (Nayak et al., 2022). As an effective tool for dissecting the genetic architecture of complex quantitative traits, genome-wide association studies provide a high-resolution approach for the identification of Quantitative trait loci (QTLs) (Sukumaran et al., 2015) and have been widely used for QTL detection of different agronomic traits in crops under different environmental conditions including seed yield related traits (Zhang et al., 2020b; Xu et al., 2018; Xiao et al., 2016), leaf architecture (Tian et al., 2011), stalk lodging-related traits (Zhang et al., 2018), and seedling root development (Pace et al., 2015). Currently, genome-wide association study (GWAS) analysis of yield-related traits in plants has been reported. Zhou et al. (2020) conducted GWAS on yield traits of 93 Bromus inermis Leyss, and a total of 95,708 effective single-nucleotide polymorphisms (SNPs) were identified, and further analysis identified 247 core SNPs related to seed yield. Akram et al. (2021) conducted GWAS on yield traits of bread wheat and found that, chromosomes 1B and 2A carried loci linked to yield in two different seasons, and an increase of up to 8.20% is possible in yield by positive allele mining. These studies give different perspectives for other gramineous plants, and provide references for genetic research of gramineous plants that have no completed whole genome sequence. In recent years, many scholars have conducted GWAS analysis on yield traits of Triticeae crops, but there is little known about natural genetic variation of forage plant seed yield. In depth analysis of the natural genetic variation and regulatory network of seed yield traits is of great significance for directional improvement of forage seed yield through molecular breeding.

Therefore, in this study, 300 germplasms from different regions were collected for the main breeding target traits of P. juncea, and a large number of SNPs covering the whole genome of P. juncea were developed by using specific locus amplified fragment sequencing (SLAF-seq) simplified genome sequencing technology. Through the two years and two locations phenotypic identification of eight seed yield related traits, the interannual growth dynamics of perennial grasses were studied, and the effects of genotypic and environmental interaction on seed yield traits of P. juncea were analyzed. The genetic basis of these eight traits was analyzed by GWAS and significant allele variants and candidate genes of target traits were explored, which laid a foundation for the research on molecular breeding of P. juncea, localization and cloning of important genes.

Materials & Methods

Plant materials and experimental design

The germplasm of P. juncea used in this study was from the China National Medium-term Gene Bank for Forage Germplasm and the United States National Plant Germplasm Resources Conservation System (NPGS). A total of 300 genotypes comprised of 45 China, 41 U.S., 43 Mongolia, 67 Russian, 15 Canada, 15 Estonia, 43 Kazakhstan and 31 former Soviet Union origin were used in the GWAS analyses. The germplasm information including sample ID, sample number, origin and cultivation are shown in Table S1. The P. juncea accessions were grown and measured under four environments (2 locations × 2 consecutive growth years), i.e., Hohhot (40°48N, 111°41E) and Baotou (40°39N, 109°49E) of Inner Mongolia China in 2021 and 2022 (Hohhot and Baotou referred to as E1 and E2 in 2021, Hohhot and Baotou referred to as E3 and E4 in 2022). A randomized block design was adopted for P. juncea planting in the field, with a row spacing of 60 cm and a plant spacing of 50 cm. Field management was consistent with local field production management. During the growth period, conventional water and fertilizer management measures were adopted, and fertilization was carried out at the tillering stage every year, with the amount of fertilization of 150 kg/hm2 18-18-18 (N-P2O5-K2O). Watering after fertilization, weeding and other field management during the growth period. Soil samples were taken from two locations and repeated three times at each location to determine soil pH, soil organic carbon (g kg−1), total nitrogen (g kg−1), soil available phosphorus (mg kg−1) and rapidly available potassium (mg kg−1). Environmental and soil conditions for each location are listed (Table S2).

Seed-yield trait measurements

The spikes of each line were harvested after full maturity. Five consistent-growth spikes were selected for seed yield related traits measurements after thorough drying in each replication. Seed yield traits including reproductive tiller length (RTL, cm), reproductive tiller number (RTN), spike length (SL, cm), spike width (SW, cm), spike node number (SNN), thousand kernel weight (TKW, g), seed weight per spike (SWS, g) and seed yield per plant (SY, g) were measured. All of the traits were represented by the mean values of the selected five spikes except RTL, RTN and SY. TKW was the average weight of three repeated measures of 1,000 randomly selected kernels from the bulked kernels of each line.

For a single environment, the mean value from five replications of each line was calculated as phenotypic data for descriptive statistics. Origin 2019b (MicroCal) was used for analysis of variance, and the significance level was 0.05. SAS 9.4 (SAS Institute, Cary, NC, USA) was used to analyze the mean, standard deviation (SD), minimum, maximum, coefficient of variance (CV), skewness and kurtosis. Pearson correlation coefficients (r) for each pair of traits was calculated using the “Hmisc” and “corrplot” packages in R software. The best linear unbiased prediction (BLUP) value of eight traits under different environments was calculated and optimized by IciMapping V4.0 (http://www.isbreeding.net) (Li, Ye & Wang, 2007). The R software package “lem4” was used to calculate the generalized heritability (h2) of seed yield traits in multiple environments. The h2 was calculated as follows: h2=Vg/Vg+Ve, where Vg represents genotype variance and Ve represents environment variance (Smith et al., 1998).

DNA extraction and SNP tag analysis

A total of 300 individual plants were selected from P. juncea materials. DNA was extracted from the P. juncea leaves of 2-3 young two-week-old seedlings using Plant Genomic DNA Kit (TIANGEN Biotech Co., Ltd, Beijing, China) following the manufacturer’s instructions. The NanoDropTM2000 spectrophotometer (Thermo Fisher Scientific, Waltham, MA, USA) was used to evaluate the quantity and quality of DNA on 1% agarose gel.

The RsaI enzyme was selected to digest the genomic DNA of each sample. The enzymatic fragments (SLAF tags) were end-repaired, ploy A tails were added, sequencing connectors were added, purification was performed, polymerase chain reaction (PCR) amplification was performed, and the target fragments were selected by mixing and gel cutting to construct libraries. Then, the libraries were tested for insert size and sequenced on the Illumina Hi Seq TM2500 sequencing platform after passing the library quality check. The raw data obtained from sequencing were identified using dual-index, and after filtering the junctions of sequenced reads, sequencing quality and data volume were evaluated (Li & Durbin, 2010). The SLAF-seq sequencing technology was used to simplify the genome sequencing for all materials of P. juncea. The sequence type with the deepest sequence in each SLAF tag was used as the reference sequence. The sequences obtained from sequencing were compared to the reference genome using BWA software (Li et al., 2009). Bread wheat (Triticum aestivum, 17 Gb) was selected as the reference genome for SNP localization (International Wheat Genome Sequencing Consortium (IWGSC), 2014). SNP markers were developed using GATK and SAMtools methods (Takeda & Matsuoka, 2008; Heffner, Sorrells & Jannink, 2009). The SNP marker intersections obtained by the methods were used as the final reliable SNP marker dataset, and then, after screening based on MAF > 0.05 and integrity > 0.85, the screened high-quality SNP loci were used for population polymorphism analysis, genetic evolution analysis, and genome-wide association analysis.

Linkage disequilibrium (LD), population genetic evolution, and kinship analysis

Genetic evolutionary relationships and environmental adaptation mechanisms were evaluated among different populations. The CM plot package in R was used for SNP marker density mapping. In this study, Gemma software (Zhou & Stephens, 2012) was used to assess the genetic relationships of populations, and then principal component analysis (PCA) was performed by Genome-wide Complex Trait (GCTA) software (Yang et al., 2011). After that, the population structure of the P. juncea natural population was analyzed by cross-validation method using Admixture 1.23 software (Montana & Hoggart, 2007), and a variety of materials in the natural population were clustered into corresponding subgroups. For the research population, the range of subgroups (K value) was set from 1 to 10, and K for iterative operations was calculated from two. The number of runs and repetitions was set to 10,000, and the number of optimal clusters was determined according to the K value with the lowest cross-validation error rate. The PopLDdecay software (Zhang et al., 2019) was used to calculate the LD of the population. The r2 of all the significant loci combinations was selected according to the selection criteria r2 > 0.1 for the LD decay plot.

GWAS and haplotype analysis to screen candidate genes

Using the mixed linear model (MLM) approach of TASSEL V5.0 software (Bradbury et al., 2007), BLUPs for eight seed yield-related trait phenotypic data were analyzed for GWAS using population structure and kinship as covariates, and the selection thresholds for significance p ≤ 1.27 × 10−6 (−log10 (p) = 5.89) and p ≤ 1.27 × 10−7 (−log10 (p) = 6.89) (Li et al., 2018b). The calculation model for BLUE value is as follows: Yikm = μ + gi + τk + gτik + δ(k)m + ɛikm, μ is the overall average, gi is the genotype effect, τk is the environmental effect, gτik is the genotype environment interaction effect, δ(k)m is the m th repeated effect within the k-th environment, and ɛikm is the random error effect. Manhattan plots were plotted using the R package CMplot. Significant differences in phenotypic traits corresponding to alleles of significantly associated markers were tested in the R program using t-tests. The screened significant markers were physical map by MapChart software to determine the physical location of the genes on the chromosomes, and the Haploview V4.2 software (Barrett, 2009) was used for haplotype analysis of loci closely associated with the target traits. Finally, the candidate genes that might be associated with the target traits were analyzed based on the genomic annotations information.

Results

Phenotypic analysis of eight seed yield related traits

Statistical analysis of seed yield revealed that there were significant phenotypic variations among 300 P. juncea germplasm in different environments and years. From 2021 to 2022, coefficient of variation for seed yield (SY) was highest (72.41%), while RTL showed the lowest CV (14.27%). By observing the data for two consecutive years, it was found that except for SL, SW and SNN, the coefficient of variation of the other five traits in 2022 was higher than that in 2021, which indicated that these traits were greatly affected by growth years (Table 1). Further analysis of variance on seed yield showed that all traits except RTN in 2022 were significantly differences between the two locations in the same year. SL, SNN, TKW and SWS were significantly different between different locations and years (Fig. 1). The joint ANOVA analysis of seed yield-related traits of P. juncea at the two experimental sites in two years showed that significant environment and genotype effects, and significant year by environment interactions were noted in RTN, SW, SNN, TKW, SWS and SY traits (P < 0.001). We calculated h2 for 300 genotypes at both locations, ranging from 0.57 (SY) to 0.81 (TKW), with an average of 0.68 (Table 2). Because of the high heritability in each environment, the least squares mean of individual traits can be calculated and used to analyze the association between markers and traits.

Table 1 Statistical descriptions of eight seed -related traits in the 300 P. juncea accessions evaluated in four environments.

Trait	Environment	Mean ± SD	Min	Max	CV/%	
RTL	2021-Hohhot	108.80 ± 16.44	59.00	143.00	15.11	
2022-Hohhot	107.94 ± 17.48	57.00	155.00	16.20	
2021-Baotou	118.44 ± 16.59	50.00	154.00	11.01	
2022-Baotou	120.47 ± 14.21	90.00	170.00	14.79	
RTN	2021-Hohhot	44.61 ± 39.64	1.00	209.00	58.87	
2022-Hohhot	52.18 ± 50.48	1.00	207.00	66.73	
2021-Baotou	123.29 ± 80.5	1.00	335.00	65.29	
2022-Baotou	61.12 ± 58.49	1.00	293.00	65.71	
SL	2021-Hohhot	11.93 ± 2.03	6.12	17.86	16.99	
2022-Hohhot	10.92 ± 5.89	5.80	15.40	13.93	
2021-Baotou	11.02 ± 1.66	6.74	16.60	16.10	
2022-Baotou	11.66 ± 1.97	6.75	18.00	15.92	
SW	2021-Hohhot	0.65 ± 0.12	0.39	1.08	19.17	
2022-Hohhot	0.58 ± 0.09	0.38	0.85	15.24	
2021-Baotou	0.68 ± 0.16	0.40	1.18	23.20	
2022-Baotou	0.66 ± 0.11	0.40	1.02	17.13	
SNN	2021-Hohhot	29.02 ± 5.97	14.00	46.00	20.56	
2022-Hohhot	30.62 ± 4.83	15.00	44.00	15.76	
2021-Baotou	30.58 ± 5.14	18.00	48.00	16.82	
2022-Baotou	32.98 ± 5.06	21.00	46.00	15.33	
TKW	2021-Hohhot	2.23 ± 0.25	0.94	3.93	26.58	
2022-Hohhot	1.89 ± 0.15	0.83	3.54	28.48	
2021-Baotou	1.95 ± 0.15	0.65	3.44	25.22	
2022-Baotou	2.29 ± 0.26	0.61	4.42	27.45	
SWS	2021-Hohhot	0.28 ± 0.11	0.11	0.69	39.81	
2022-Hohhot	0.22 ± 0.09	0.07	0.53	40.85	
2021-Baotou	0.23 ± 0.09	0.09	0.56	40.42	
2022-Baotou	0.34 ± 0.15	0.10	0.96	42.60	
SY	2021-Hohhot	13.38 ± 13.58	0.18	77.56	72.51	
2022-Hohhot	11.77 ± 12.25	0.13	74.57	74.09	
2021-Baotou	30.69 ± 22.04	0.12	122.12	71.21	
2022-Baotou	21.11 ± 21.43	0.13	97.89	71.81	
Notes.

The eight traits included reproductive tiller length (RTL), reproductive tiller number (RTN), spike length (SL), spike width (SW), spike node number (SNN), thousand kernel weight (TKW), seed weight per spike (SWS), seed yield per plant (SY). Hohhot and Baotou were the two locations used for field evaluation of the traits.

The correlation coefficients between eight seed yield related traits were analyzed. These eight indicators were normally distributed or skewed. Pearson’s correlation coefficients indicated that there were five agronomic traits with extremely significant correlation with SY in Hohhot (P < 0.001), seven agronomic traits with extremely significant correlation with SY in Baotou, and the correlation coefficient of RTN was the largest. SNN was correlated with the six agronomic traits, among which SL had the strongest correlation (Fig. 2). This suggests that the traits are genetically linked or that the traits are affected by genes with pleiotropic effects. It also indicates that there is a significant interaction between seed-related traits, and the improvement of RTN has a significant effect on seed yield.

Figure 1 Boxplots of the mean amounts of eight seed yield-related traits of different locations during 2021–2022.

*p < 0.05; different capital letters indicate the difference significance between the same place in different years.

Table 2 ANOVA of seed yield-related traits evaluated at two locations for two years.

Source	DF	RTL	RTN	SL	SW	SNN	TKW	SWS	SY	
Environment (E)	1	180.47***	50,068***	1.5NS	653.44***	411.83***	282.53***	444.17***	30,429.3***	
Year (Y)	1	1.33NS	2,075***	6.93**	276.85***	401.48***	85.13***	325.33***	5,414.58***	
Genotype (G)	299	2.52***	684.65***	6.75***	15.34***	23.47***	48.7***	26.79***	473.77***	
E×Y	1	1.26NS	32,874***	197.71***	127.71***	14.87***	1,686.57***	1,982.99***	12,863***	
G×E	299	1.13NS	315.72***	2.8***	7.99***	5.73***	16.65***	11.41***	247.02***	
G×Y	299	1.3**	195.08***	1.94***	7.6***	5.79***	14.71***	8.88***	194.4***	
G×E×Y	299	0.88NS	177.27***	1.91***	3.19***	5.95***	16.34***	9.08***	4.76***	
Error	1,069	3.78	12.40	1.23	8.57	4.21	3.54	4.43	10.32	
Heritability (h2)	–	0.75	0.65	0.68	0.62	0.69	0.81	0.64	0.57	
Notes.

NS, Non-significant; DF, Degree of freedom.

*, ** and *** indicate 0.05, 0.01, and 0.001 levels of significance.

Figure 2 Pearson correlation coefficients for the eight seed yield-related traits evaluated in the two locations.

(A) Hohhot, (B) Baotou. Statistical significance of coefficients labeled as ***P < 0.0001, **P < 0.01, *P < 0.05.

Development of SNP markers, population structure and linkage disequilibrium analysis of P. juncea

The SLAF-seq technology was used to simplify the genome sequencing of 300 P. juncea germplasm, and genome-wide SNP markers of P. juncea were developed and obtained. Using the 1,176,240 SLAF tags obtained, a total of 84,024 SNPs’ accurate positioning on the bread wheat genome was established based on the criteria integrity > 0.85 and MAF > 0.05, including 19,913 on subgenome A, 24,760 on subgenome B, and 39,351 on subgenome D, accounting for 23.7%, 29.5% and 46.8% of the total number of SNPs, respectively. The chromosomal level distribution of the 84,024 SNPs on the three subgenomes showed the maximum SNPs in Chr2D (6,717), while the minimum SNPs in Chr6A (2,141). In general, SNP markers are evenly distributed on the genome of P. juncea, which provides an important guarantee for genetic diversity analysis and genome-wide association analysis of P. juncea (Table 3).

Table 3 Chromosome wise-distribution and genetic diversity of 84,024 SNPs and the intra-chromosomal estimated LD among 300 P. juncea genotypes.

Chr	No. of SNP	D prime	No of marker pairs in perfect LD	
1A	2,381	0.848	5,523	
1B	2,994	0.594	7,083	
1D	4,858	0.537	16,833	
2A	3613	0.560	12542	
2B	3,961	0.541	8,465	
2D	6,717	0.367	36,193	
3A	3,234	0.844	8,384	
3B	4,282	0.531	11,137	
3D	6,122	0.380	23,018	
4A	2,604	0.857	5,275	
4B	3,149	0.891	8,341	
4D	5,051	0.625	20,224	
5A	2,855	0.899	6,034	
5B	3,753	0.571	9,690	
5D	5,868	0.328	25,089	
6A	2,141	0.810	5,035	
6B	2,794	0.881	6,476	
6D	4,330	0.429	18,403	
7A	3,085	0.591	8,463	
7B	3,827	0.826	9,329	
7D	6,405	0.397	23,211	
A subgenome	19,913	0.773	7,322	
B subgenome	24,760	0.691	8,646	
D subgenome	39,351	0.438	23,282	

Population structure indicates the degree of genetic variation between individuals or populations. Population structure was analyzed using 84,024 SNPs, and the combination trends of the likelihood values of LnP(D) and ΔK calculated for each K showed that the 300 genotypes could be assigned into five subgroups. Based on the Q value, it can be found that five subgroups were clustered according to geographical sources, including 44 (Former Soviet Union and Estonia), 72 (Russia), 84 (China and Mongolia), 41 (Kazakhstan) and 59 (U.S. and Canada) lines, respectively (Figs. 3A, 3D). Three hundred P. juncea genotypes were evaluated using PCA, and PC scatter plots showed that the first and second PC were composed of subgroups of different geographical sources (Fig. 3B). The seed yield traits of the five subgroups were analyzed. Group 1 had the highest values for all traits, while Group 4 had the lowest values, and the seed yield traits of the other three groups were not significantly different (Table S3).

Figure 3 LD decay, PCA, and population structure of P. juncea.

(A) LnP(D), the log probability of the data. K means subpopulations. (B) Principal component analysis (PCA) of 300 accessions. (C) LD decay for five subgroup. The values on the Y-axis represent the squared correlation coefficient (r2) and the values at X-axis represent physical distance (kb). (D) Population structure. Bar graphs for five subpopulations are indicated by different colors. The vertical coordinates of each subpopulation indicate the membership coefficient for each individual, and the digits on the horizontal coordinates represent the corresponding genotypes corresponding to the table. In each subpopulation, each vertical bar represents one genotype.

Chromosome wise LD plots were generated using 84,024 selected SNP markers (Fig. 3C). The complete statistics of the 84,024 markers, including SNP numbers, chromosome LD and markers with perfect LD are presented in Table 3. The results indicated that there are different levels of LD on different chromosomes of each subgenome. We found that LD decays rate is different in different subgroups, with the fastest decay rate is Former Soviet Union and Estonia, and the slowest is Kazakhstan.

GWAS, pleiotropism SNPs screening and favorable allele variation

We further performed GWAS on the differences in eight seed yield-related traits to identify significantly associated genes. A total of 121 SNPs significantly associated with the target traits were detected on all 21 chromosomes of wheat using 84,024 SNPs associated with eight seed yield phenotype traits and BLUP values in four environments, respectively. And explained (R2) between 4.15% and 15.64% of the variation. Notably, seven, 24, 48, seven, six, nine, 14 and six significantly associated SNPs were obtained for RTL, RTN, SL, SW, SNN, TKW, SWS and SY, respectively (Fig. 4). Chromosome wise, the highest number of SNPs was detected on Chr2D (12 SNPs), followed by Chr5A, 5D and 7D (nine SNPs). Nineteen SNPs were found to be associated with the target trait in four different environments, with the lowest P-value (Table S4).

Figure 4 Manhattan plots for eight different traits of P. juncea.

The ring from inside to outside denotes RTL, RTN, SL, SW, SNN, TKW, SWS and SY respectively. The red horizontal line represents the draw P value threshold (P value < 1.27E−06). The blue dash line indicates Bonferroni corrected P value (P value < 1.27E−07). The outermost circle is the SNP density map.

We observed that fifteen SNPs were associated with more than one seed yield related traits. Among them, Chr7A_675095747 was related to four traits of RTN, SL, SNN, and TKW, which was an important locus for regulating seed yield. At the same time, this locus was detected in multiple environments, and the P-value showed extremely significant. SL, RTN and SY were associated with six, five and five pleiotropism loci respectively. Besides multi-trait SNPs, some genomic regions contained multi-QTLs for more than one trait. The genomic regions harboring multi-SNPs for more than one trait included Chr2A (72.6–88.7 Mb), which had three SNPs associated with seed yield related traits; Chr2D (111.6–142.4Mb, 637.7–650.9Mb), with eight SNPs, four associated with SL; Chr7D (6.8–32.6 Mb, 110.6–146.4Mb) had four SNPs associated with RTN, SL, SNN and SY (Fig. 5).

Figure 5 Distribution of 121 significantly associated SNPs on 21 chromosomes based on the physical distance from the eight seed yield agronomic trait differences.

The numbers on the right side of each column represent the physical location (Mb) of each lead SNP. Letters to the left of each column represent the SNP.

In order to study the phenotype effects of significantly associated SNP locus allele variations, eight pleiotropism loci that could be detected in four different environments were selected, and the phenotype traits corresponding to genotype were analyzed. The allele variation of each SNP locus was Chr2A_72607643 (G/A), Chr2D_640215049 (G/T), Chr2D_650894588 (G/A), Chr4A_722894581 (C/T), Chr5A_657544633 (C/G), Chr5D_405156269(C/T), Chr6A_17541092 (G/A) and Chr7A_675095747 (G/T). Further analysis showed that Chr2A_72607643-AA, Chr2D_640215049-TT, Chr2D_650894588-AA, Chr4A_722894581-CC, Chr5A_657544633-GG Chr5D_405156269-TT, Chr6A_17541092-AG and Chr7A_675095747-GG were favorable allele variations for seed yield-related traits of P. juncea (Fig. S1). To further determine whether the favorable allele variation significant associated with seed yield-related traits have a polymerization effect, we compared and analyzed phenotypic data of P. juncea with different numbers of favorable allele variation loci. The results showed that among the eight significantly related SNPs, each germplasm contained up to five favorable allele variation and at least one favorable allele variation. Compared with materials with less favorable allele variation loci, the plant with more favorable allele variation loci showed excellent traits. For example, the average SY value of P. juncea containing one favorable allele variation is 14.91 g, while the average SY value of P. juncea containing five favorable allele variation is 19.47 g. These results indicate that favorable allele variation loci have a significant polymerization effect on eight seed yield traits (Fig. S2).

Candidate gene annotation and haplotype analysis

In addition, 19 significant SNPs with the smallest P-values for each seed yield trait and detected in four different environments were selected, and the predicted candidate genes were identified in the range of 100 kb upstream and downstream of the SNP. A total of 91 candidate genes were obtained from 19 SNPs within the candidate regions (Table 4). Two candidate genes were detected for RTL, including TraesCS2A02G122600.1 encoding a cellulose synthase-like protein (CSLD) on chromosome 2A within 83.85 kb of an SNP, TraesCS2A02G122700.1 encoding a momilactone A synthase. A total of three SNP significantly associated with RTN were detected and 13 candidate genes were annotated. TraesCS2D02G577000.1 encoding a serine/threonine-protein kinase (STK) on chromosome 2D within 82.43 kb of target SNP. Four candidate genes, TraesCS4A02G457400.1, TraesCS4A02G457800.1, TraesCS4A02G457500.1 and TraesCS4A02G457600.3 on chromosome 4A, which were about 54.79 kb, 10.49 kb, 54.79 kb and 7.2 kb away from the target SNP, respectively. TraesCS7A02G483300.1 encoding an F-box protein within 72.13 kb of target SNP. A total of six SNP significantly associated with SL were detected and 30 candidate genes were annotated. TraesCS2D02G599600.1 encoding mevalonate kinase (MK), TraesCS2D02G599800.1 encoding LRR receptor-like STK, TraesCS2D02G599900.1 encoding CBS domain-containing protein (CDCP), TraesCS2D02G600000.2 encoding callose synthase, TraesCS2D02G600100.1 encoding stem-specific protein (TSJT1) on chromosome 2D. These genes were located approximately 30.87-, 11.06-, 32.99-, 37.26- and 61.02- from the target SNPs, respectively. Four candidate genes, TraesCS5A02G486900.1, TraesCS5A02G487000.1, TraesCS5A02G487300.1 and TraesCS5A02G487500.1 on chromosome 5A, encoding photosystem II reaction center protein, STK, cytochrome P450 and F-box protein, respectively. Three candidate genes, on chromosome 5D, annotation information is RNA polymerase II and putative pentatricopeptide repeat-containing protein (PPR). Only one of the four candidate genes of Chr6A_17541092 was annotated to transcription repressor OFP14. One candidate gene TraesCS7B02G250800.1 encoding a GDSL esterase/lipase on chromosome 7B within 1.6 kb of an SNP. A total of 16 candidate genes related to SW were annotated, among which, three genes encoding membrane protein of ER body-like protein (MEBL), three genes encoding STK. One candidate gene, TraesCS7A02G483300.1 on chromosome 7A was detected for SNN and TKW, encoding an F-box protein at 72.13 kb from the target SNP. In addition, three candidate genes of Chr2D _ 640215049, which were significantly associated with TKW, were detected, including TraesCS2D02G576800.1 encoding a DDB1- and CUL4-associated factor homolog (DCAF) and TraesCS2D02G576900.1 encoding uncharacterized acetyltransferase. A total of four SNP were significantly associated with SWS and SY, and 17 candidate genes were annotated. These four SNPs were all pleiotropism loci, and the significant associated SNPs of SL and RTL are the same loci with the same annotation information.

Table 4 The potential candidate genes identified corresponding to the significant SNP associated with eight seed yield traits in the population.

Traits	SNP	Gene ID	Start	End	Distance to peak SNP	Encoding Protein	
RTL	Chr2A_72607643	TraesCS2A02G122600.1	72691495	72695638	5′_83852	Cellulose synthase-like protein	
TraesCS2A02G122700.1	72696518	72697360	3′_88875	Momilactone A synthase	
RTN	Chr2D_640215049	TraesCS2D02G577000.1	640297481	640299386	3′_82432	Serine/threonine-protein kinase SAPK5	
Chr4A_722894581	TraesCS4A02G457400.1	722837844	722839786	3′_54795	30S ribosomal protein S13, RPS13	
TraesCS4A02G457500.1	722841516	722845182	5′_49399	Glutaminyl-peptide cyclotransferase	
TraesCS4A02G457600.3	722882391	722887381	3′_7200	Membrane protein of ER body-like protein, MEBL	
TraesCS4A02G457700.2	722899708	722905412	5′_5127	Membrane protein of ER body-like protein, MEBL	
TraesCS4A02G457700.1	722899708	722905534	5′_5127	Membrane protein of ER body-like protein, MEBL	
TraesCS4A02G457800.1	722905079	722906903	3′_10498	30S ribosomal protein S13, RPS13	
TraesCS4A02G457900.1	722907610	722916249	3′_13029	Disease resistance protein RPM1	
Chr7A_675095747	TraesCS7A02G483300.1	675021874	675023612	3′_72135	F-box protein	
TraesCS7A02G483400.1	675025492	675027295	3′_68452	–	
TraesCS7A02G483400.2	675025492	675027309	3′_68438	–	
TraesCS7A02G483500.1	675046750	675047270	3′_48477	–	
TraesCS7A02G483600.1	675112899	675115741	3′_17152	–	
SL	Chr2D_650894588	TraesCS2D02G599500.1	650831742	650837613	5′_56975	–	
TraesCS2D02G599600.1	650860731	650863713	5′_30875	Mevalonate kinase	
TraesCS2D02G599700.1	650869233	650871862	5′_22726	–	
TraesCS2D02G599800.1	650879896	650883522	5′_11066	Probable LRR receptor-like serine/threonine-protein kinase	
TraesCS2D02G599900.1	650927579	650929016	5′_32991	CBS domain-containing protein	
TraesCS2D02G600000.2	650931852	650947979	5′_37264	Callose synthase	
TraesCS2D02G600100.1	650955614	650957060	5′_61026	Stem-specific protein TSJT1	
TraesCS2D02G600200.1	650962449	650964288	5′_67861	E3 ubiquitin-protein ligase SINA-like 2	
Chr5A_657544633	TraesCS5A02G486900.1	657452611	657452823	3′_91810	Photosystem II reaction center protein	
TraesCS5A02G487000.1	657481959	657486595	3′_58038	Probable serine/threonine-protein kinase	
TraesCS5A02G487000.2	657481959	657486706	3′_57927	Probable serine/threonine-protein kinase	
TraesCS5A02G487000.3	657483607	657486595	3′_58038	Probable serine/threonine-protein kinase	
TraesCS5A02G487100.1	657490411	657493777	3′_50856	–	
TraesCS5A02G487200.1	657496902	657498825	5′_45808	–	
TraesCS5A02G487300.1	657537556	657539589	3′_5044	Cytochrome P450	
TraesCS5A02G487400.1	657540736	657548288	–	–	
TraesCS5A02G487500.1	657550034	657554787	5′_5401	F-box protein	
Chr5D_405156269	TraesCS5D02G308200.1	405154704	405157124	–	RNA polymerase II transcriptional coactivator KIWI	
TraesCS5D02G308300.1	405209676	405211820	3′_53407	Putative pentatricopeptide repeat-containing protein	
TraesCS5D02G308400.1	405251408	405255247	3′_95139	–	
Chr6A_17541092	TraesCS6A02G035500.1	17528410	17531735	3′_9357	–	
TraesCS6A02G035600.1	17540023	17543295	–	–	
TraesCS6A02G035700.1	17601604	17602335	5′_60512	Transcription repressor OFP14	
TraesCS6A02G035800.1	17638201	17641019	3′_97109	–	
Chr7A_675095747	TraesCS7A02G483300.1	675021874	675023612	3′_72135	F-box protein	
TraesCS7A02G483400.1	675025492	675027295	3′_68452	–	
TraesCS7A02G483400.2	675025492	675027309	3′_68438	–	
TraesCS7A02G483500.1	675046750	675047270	3′_48477	–	
TraesCS7A02G483600.1	675112899	675115741	3′_17152	–	
Chr7B_463032072	TraesCS7B02G250800.1	463033674	463037960	3′_1602	GDSL esterase/lipase EXL1	
SW	Chr4A_722894581	TraesCS4A02G457400.1	722837844	722839786	3′_54795	30S ribosomal protein S13, RPS13	
TraesCS4A02G457500.1	722841516	722845182	5′_49399	Glutaminyl-peptide cyclotransferase	
TraesCS4A02G457600.3	722882391	722887381	3′_7200	Membrane protein of ER body-like protein, MEBL	
TraesCS4A02G457700.2	722899708	722905412	5′_5127	Membrane protein of ER body-like protein, MEBL	
TraesCS4A02G457700.1	722899708	722905534	5′_5127	Membrane protein of ER body-like protein, MEBL	
TraesCS4A02G457800.1	722905079	722906903	3′_10498	30S ribosomal protein S13, RPS13	
TraesCS4A02G457900.1	722907610	722916249	3′_13029	Disease resistance protein RPM1	
Chr5A_657544633	TraesCS5A02G486900.1	657452611	657452823	3′_91810	Photosystem II reaction center protein	
TraesCS5A02G487000.1	657481959	657486595	3′_58038	Probable serine/threonine-protein kinase	
TraesCS5A02G487000.2	657481959	657486706	3′_57927	Probable serine/threonine-protein kinase	
TraesCS5A02G487000.3	657483607	657486595	3′_58038	Probable serine/threonine-protein kinase	
TraesCS5A02G487100.1	657490411	657493777	3′_50856	–	
TraesCS5A02G487200.1	657496902	657498825	5′_45808	–	
TraesCS5A02G487300.1	657537556	657539589	3′_5044	Cytochrome P450	
TraesCS5A02G487400.1	657540736	657548288	–	–	
TraesCS5A02G487500.1	657550034	657554787	5′_5401	F-box protein	
SNN	Chr7A_675095747	TraesCS7A02G483300.1	675021874	675023612	3′_72135	F-box protein	
TraesCS7A02G483400.1	675025492	675027295	3′_68452	–	
TraesCS7A02G483400.2	675025492	675027309	3′_68438	–	
TraesCS7A02G483500.1	675046750	675047270	3′_48477	–	
TraesCS7A02G483600.1	675112899	675115741	3′_17152	–	
TKW	Chr2D_640215049	TraesCS2D02G576700.1	640174317	640176964	3′_38085	–	
TraesCS2D02G576800.1	640206750	640216632	–	DDB1- and CUL4-associated factor homolog 1 GN=DCAF1	
TraesCS2D02G576900.1	640240670	640242328	5′_25621	Uncharacterized acetyltransferase	
Chr7A_675095747	TraesCS7A02G483300.1	675021874	675023612	3′_72135	F-box protein	
TraesCS7A02G483400.1	675025492	675027295	3′_68452	–	
TraesCS7A02G483400.2	675025492	675027309	3′_68438	–	
TraesCS7A02G483500.1	675046750	675047270	3′_48477	–	
TraesCS7A02G483600.1	675112899	675115741	3′_17152	–	
SWS	Chr2A_72607643	TraesCS2A02G122600.1	72691495	72695638	5′_83852	Cellulose synthase-like protein D3	
TraesCS2A02G122700.1	72696518	72697360	3′_88875	Momilactone A synthase	
Chr5D_405156269	TraesCS5D02G308200.1	405154704	405157124	–	RNA polymerase II transcriptional coactivator KIWI	
TraesCS5D02G308300.1	405209676	405211820	3′_53407	Putative pentatricopeptide repeat-containing protein	
TraesCS5D02G308400.1	405251408	405255247	3′_95139	–	
SY	Chr2D_650894588	TraesCS2D02G599500.1	650831742	650837613	5′_56975	–	
TraesCS2D02G599600.1	650860731	650863713	5′_30875	Mevalonate kinase	
TraesCS2D02G599700.1	650869233	650871862	5′_22726	–	
TraesCS2D02G599800.1	650879896	650883522	5′_11066	Probable LRR receptor-like serine/threonine-protein kinase	
TraesCS2D02G599900.1	650927579	650929016	5′_32991	CBS domain-containing protein CBSX5	
TraesCS2D02G600000.2	650931852	650947979	5′_37264	Callose synthase 7	
TraesCS2D02G600100.1	650955614	650957060	5′_61026	Stem-specific protein TSJT1	
TraesCS2D02G600200.1	650962449	650964288	5′_67861	E3 ubiquitin-protein ligase SINA-like 2	
Chr6A_17541092	TraesCS6A02G035500.1	17528410	17531735	3′_9357	–	
TraesCS6A02G035600.1	17540023	17543295	–	–	
TraesCS6A02G035700.1	17601604	17602335	5′_60512	Transcription repressor OFP14	
TraesCS6A02G035800.1	17638201	17641019	3′_97109	–	

To further study the role of candidate genes, we conducted haplotype analyses on the 300 P. juncea genotypes to identify the elite haplotypes. We analyzed the four most significant SNPs among the nineteen SNPs that were consistently present in four environments. Chr2D_640215049, which was significantly associated with RTN and TKW, formed a 0 kb linkage region with the surrounding six SNPs (Fig. 6A). A total of five haplotypes were found in this region. RTN and TKW containing Hap.4 were higher than other haplotypes, indicating that Hap.4 was an excellent haplotype (Figs. 6B, 6C). Chr2D_650894588, which was significantly associated with SL and SY, formed a 46 kb linkage region with the surrounding two SNPs (Fig. 7A). This linkage region contains five haplotypes, and it was found that SL and SY containing Hap.4 material were higher than other haplotype materials (Figs. 7B, 7C). Similarly, the Chr6A_17541092 significant association with SL and SY was detected to form a 0 kb linkage region with four SNPs (Fig. 8A). This linkage region contains four haplotypes, and it was found that SL and SY containing Hap.3 material were higher than other haplotype materials (Figs. 8B, 8C). The Chr7A_675095747 significant association with RTN, SL, SNN and TKW was detected to form a 0 kb linkage region with two SNPs (Fig. 9A). This linkage region contains five haplotypes, and it was found that the average value of each trait of containing the Hap.4 was higher than other haplotype. The haplotype analysis suggests that Chr7A_675095747 Hap.4 is strongly associated with seed yield in P. juncea (Figs. 9B, 9C).

Figure 6 Haplotype analysis of Chr2D_640215049 and phenotypes of RTN and TKW among different haplotypes.

(A) Associated linkage regions. (B) Haplotype. (C) Phenotype. Num., number of accessions.

Figure 7 Haplotype analysis of Chr2D_650894588 and phenotypes of SL and SY among different haplotypes.

(A) Associated linkage regions. (B) Haplotype. (C) Phenotype. Num., number of accessions.

Figure 8 Haplotype analysis of Chr6A_17541092 and phenotypes of SL and SY among different haplotypes.

(A) Associated linkage regions. (B) Haplotype. (C) Phenotype. Num., number of accessions.

Figure 9 Haplotype analysis of Chr7A_675095747 and phenotypes of RTN SL SNN and TKW among different haplotypes.

(A) Associated linkage regions. (B) Haplotype. (C) Phenotype. Num., number of accessions.

Discussion

Trait variations and correlations in P. juncea

P. juncea is one of the most important perennial grasses in the high latitude areas of the Northern Hemisphere, with economic, feeding and breeding research value. Breeding P. juncea germplasm that can adapt to different regions is particularly important for efficient utilization and production (Cuevas & Prom, 2020). From previous research results, we selected eight quantitative traits related to seed yield for measurement and analysis in two environments for two consecutive years. The results showed that from 2021 to 2022, the coefficient of variation of SY was the largest, ranging from 71.21% to 74.09%. A similar result was also found by Zhang et al. (2018) in their study on seed yield traits of P. juncea germplasm. This indicates that there was extensive phenotypic variation and rich genetic diversity in P. juncea. The variance analysis of seed yield traits of two locations showed that the seed-trait indicators in Baotou were higher than in Hohhot over the years. This could be attributed to differences in the climate conditions between the two locations. The annual effective sunshine hours, daily mean temperature and relative humidity in Baotou were significantly higher than in Hohhot. Studies have shown that these factors have a positive effect on the seed yield per plant, and the relative humidity of the air has a promoting effect on plant growth (Liang et al., 2023). The relatively high generalized heritability of all traits showed that seed traits of P. juncea are high heritability traits, and the natural population has rich genetic diversity in them. Although affected by both genotype and environmental factor, seed traits are mainly controlled by genotype due to high heritability.

Correlation analysis further revealed the majority of seed yield-related traits in the associated population were significantly positively correlated. SY had a significant correlation with multiple traits, and a highly significant positive correlation with RTN. Zhang et al. (2018) found that although the correlation between seed setting rate and seed yield per plant and its direct effects were not significant, the seed setting rate mainly played a role through the indirect effect on the number of reproductive branches and the number of spikelets per panicle. and this result was consistent with the current study. In this research, SW was significantly positively correlated with TKW, and similar finding was reported in Secale cereale subsp. Segetale (Mu et al., 2019). Therefore, we speculated that the increase of SW would increase kernels per spikelet and TKW. Spike is the primary condition for seed yield formation (Bian et al., 2020). In this research, we also found a large number of traits significantly related to SL and SW, which indirectly indicated that these traits co-regulate seed yield. At the same time, there were some differences in the adaptability to the environment, which produced genetic variations and showed different traits (Xie et al., 2015). The above results showed that there were significant genetic variations in the seed yield traits of P. juncea among different materials, and the seed traits data of P. juncea from multi-years and locations could provide reliable phenotypic data for GWAS.

Population structure and LD decay of P. juncea

The accuracy of GWAS is influenced by population structure, kinship, and LD decay distance. Thus, the MLM method was used to take these factors into account for GWAS analysis (Zhang et al., 2010). In this research, through the analysis of population genetic structure of P. juncea individual lines from different countries and regions, it was found that the population structure of 300 P. juncea individual lines had a certain correlation with geographical distribution, which was also verified through PCA. This may be attributed to the genetic background of the genotype diversity in this studied P. juncea materials, indicating that SNP markers can effectively group genotypes according to gene composition. Through different statistical methods, it was found that although both methods clustered 300 P. juncea lines into five groups, there were some individual lines was clustered differently. The main reason for this result may be due to breeding domestication, which has a significant impact on the diversity structure. In addition, different environments can also cause genetic changes, which can affect the division of population structure.

LD determines the marker density required for GWAS analysis, and is one of the important indicators to measure whether there is correlation between molecular markers. Li et al. (2023) used SSR markers to estimate a wider distribution of LD, which is difficult to estimate accurately. In this study, LD decay rate is different in subgroups, with the fastest decay rate in the population from the former Soviet Union and Estonia, indicating the high genetic diversity of the P. juncea population in this region. The above analysis indicates that the sample population in this research has high genetic diversity, suitable population structure and LD decay level, indicating that the population is suitable for GWAS. High genetic diversity generally implies the presence of more allelic variations in the population, which may increase the probability of detecting true association signals, thereby enhancing the statistical power of GWAS.

Significant SNPs and potential candidate genes

Seed yield is a super complex quantitative trait controlled by multiple genes and susceptible to environmental influences. It is the underlying reason of multiple seed yield-related traits be analyzed in the research. So far, there are few genome information of P. juncea on seed yield-related trait. Therefore, we used GWAS on the seed yield-related traits of P. juncea, and comprehensively compared with the results of previous studies on wheat genome association analysis to further determine the accuracy of SNP loci significantly associated with seed yield traits in the previous analysis.

Genome-wide association analysis showed that a total of 121 SNPs were significantly associated with eight seed yield-related traits. Among them, seven loci were significantly associated with RTL. A SNP (Chr2A_72607643) that regulates RTL was found at the position of 72.6 Mb on the short arm of Chr2A in wheat, which is consistent with the reported by Kuang et al. (2020), confirming that the gene region was involved in the regulation of RTL and two candidate genes were annotated. The annotation information of TraesCS2A02G122600.1 is CSLD. Previous studies have shown that OsCSLD4 plays an important role in cell wall polysaccharide synthesis, participates in the regulation of growth and differentiation of rice cells, and has an important effect on rice plant height and leaf type (Yoshikawa et al., 2013). OsCSLD4 also plays an important role in rice grain development. Overexpression of OsCSLD4 promotes rice seed growth and significantly widens the grains, indicating that OsCSLD4 plays an important role in synergistically promoting rice growth and increasing seed yield (Zhao et al., 2022).

Thousand grain weight, grain number per spike and number of reproductive branches significantly contribute to seed yield of P. juncea. In this study, 24 SNPs were found to be significantly associated with RTN, distributed on 14 chromosomes, and a single locus could explain 4.33%–12.44% of the phenotypic variation. Cui et al. (2014) reported a QTL (QSnpp-4A.3) located on chromosome 4A controlling RTN in wheat, was detected as the locus Chr4A_722894581 in this study, and the locus was detected in all environments. the locus could annotated to seven candidate genes, TraesCS4A02G457400.1 and TraesCS4A02G457800.1 annotation information was 30S ribosomal protein (RPS). In recent years, a large number of research reports have reported that the function of RPS in plants is mainly manifested in the change of phenotype of mutant plants. The abnormal development of female gametophyte in Arabidopsis thaliana RPL24B mutant, leading to seed abortion (Nishimura et al., 2005). The RPL23aA mutation causes slow root growth and lateral root deformity in Arabidopsis (Degenhardt & Bonham-Smith, 2008). Therefore, we speculate that the gene will regulate the reproductive development of plants, thereby affecting the number of reproductive branches of P. juncea. Cao et al. (2020) located the QTL of RTN to the position of 675 Mb on chromosome 7A, which aligned to Chr7A_675095747 in this study, and annotated to five candidate genes. TraesCS7A02G483300.1 gene encoded F-box protein. F-box protein is reported to be involved in signal transduction of plant hormones (ethylene, auxin, gibberellin, jasmonic acid), flower organ development and other biological processes (Xu et al., 2021), the haplotype of Chr7A_675095747 was analyzed and Hap.4 was found the elite haplotype. The elite haplotypes can be selected in molecular assisted selection in the breeding of P. juncea. Although the physical location of Chr2D_640215049 is different from the reported QTLs or genes related to RTN, it is detected in multiple environments, and the RTN of haplotype-AACTCTT (Hap.4) of this locus is significantly higher than that of other haplotypes, which is speculated to be a potential RTN related new locus.

Spike length is closely related to yield traits such as spikelet number, grain number per spike and thousand grain weight. Among 48 loci significantly associated with SL detected on 19 wheat chromosomes, The SNP (Chr7A_675095747) regulating RTN is also in the column, andis consistent with the SNP found in winter wheat by Gill et al. (2022). Yao (2017) found that OsFBX76- OE reduced thousand grain weight, seed setting rate and yield per plant, while OsFBX76- RNAi increased seed yield per plant. Therefore, OsFBX76 affects the yield per plant of rice by negatively regulating grain size and seed setting rate. Li (2020) found the SNP regulating SL on the chromosome 7B of wild Emmer wheat was consistent with Chr7B_463032072 found in this study, and the annotated gene TracesCS7B02G250800.1 encodes the GDSL lipase gene family, which was essential for the development of anther and pollen as well as hormone signal transduction (Zhang, 2020a). Therefore, we speculate that these genes affect the morphological changes of plants and spike by regulating hormone changes in plants. We further analyzed the other two significantly associated SNPs that regulate SL. Chr2D_650894588 annotated to the TraesCS2D02G599600.1 encodes the MK. At present, the research on MK mainly focuses on important agronomic traits such as plant overexpression and secondary metabolites, grain size and grain weight (Champenoy & Tourte, 1998). TraesCS2D02G599900.1 encodes CDCP. CDCP can improve the utilization efficiency of nitrogen in plants, so as to achieve the purpose of increasing yield (Hao et al., 2016). In this study, the significantly associated SNPs on the short arm of chromosome 6A did not overlap with the physical positions of previous QTLs of SL. However, this locus was detected in multiple environments. Therefore, it may be potential locus to be discovered. Chr6A_17541092 annotates the TraesCS6A02G035700.1 encoding the transcriptional repressor OFP14, which found regulating the grains shape of rice, and overexpression of OsOFP19 leads to phenotype of significantly reduced spike length, resulting in a decrease in seed setting rate and number of spikes (Zhao et al., 2018). Meanwhile, Chr6A_17541092 Hap.3 is an elite haplotype which have advantages on SL and SY traits.

A total of seven loci were significantly associated with SW in this study. A stable major QTL associated with SW was reported on chromosome 5A (qSW-5A.1, 646.3–657.5 Mb) (Shui et al., 2020). The SNP Chr5A_657544633 was also associated with the same position, and was annotated to nine candidate genes. The TraesCS5A02G487000.1 encoded STK. It has been found that ZmSTK1 is a protein kinase specifically expressed in maize pollen, which plays a role in the development of maize pollen or the elongation of pollen tubes, and pollination and fertilization are closely related to the growth and fruiting of maize (Fan et al., 2018). Therefore, we infer that the STK affects seed growth and compactness by affecting the pollination ability of P. juncea during flowering stage, which leads to the change of SW. The results laid the foundation for further research on spike width traits.

The number of grains per spike is the main factor affecting the seed yield of Triticeae crops. The number of grains per panicle is mainly determined by the number of spike rachis nodes. Fan et al. (2024) cloned a gene HvSRN1 that controls the number of spike rachis nodes in barley, and clarified the regulation of this gene on traits such as spike length and yield per plant in barley. QTLs regulating SNN were also found to be distributed in Chr7A (qNRN-7A.h2, 674 Mb) in wheat (Voss-Fels et al., 2019). The candidate gene TraesCS7A01G481600 encodes F-box protein. In this study, Chr7A_675095747 was similar to the locus located on chromosome 7A in previous studies and was detected in multiple environments. Therefore, it can be determined that this locus controls SNN. Further analysis identified a candidate gene TraesCS7A02G483300.1 on chromosome 7A encoding F-box protein, which is the same as the annotation information in the above study, and is homologous to the rice ABBERANT PANICLE ORGANIZATION1 (APO1) gene, which has a significant effect on spike morphological changes.

So far, extensive research has been conducted of thousand grain weight, which is essential for understanding its genetic mechanism. In this study, a total of nine SNPs related to TKW were detected. Cao et al. (2020) located the QTL of TKW in wheat at the position of 634.5–647.5 Mb (QTKW.ndsu.2D). Guan et al. (2018) QTL of TKW in wheat at the position of 675 Mb position (QTgw.cau-7A.4). They are aligned to the SNPs on chromosome 2D and 7A in this study. TraesCS2D02G576800.1 annotated by Chr2D_640215049 encode DCAF. The DCAF gene is associated with abortion and blighted grain in rice (Luo et al., 2011), leading to a decrease in seed yield. A total of fourteen SNPs were associated with SWS. The two SNPs identified on Chr5D and Chr6D aligned to reported QTLs (Liu et al., 2023). Chr5D_405156269 has annotated three candidate genes, with annotated information for RNA polymerase II and PPR, respectively. Research has shown that RNA polymerase II is involved in many biological processes, including gene regulation, cell differentiation, cell proliferation, and so on (Zhang et al., 2022). Liu et al. (2016) cloned the TaPPR4 in wheat and found that TaPPR4-OE wheat lines tiller number decreased, the spike length increased, the number of grains per spike decreased, and the seeds were shriveled. This result is of great significance for the study of PPR protein in other plants. Six SNPs were significantly associated with SY. SNP loci (AX_109941480, 643.0–650.7 Mb) controlling seed yield have been found on wheat 2D chromosome (Li et al., 2018a), which are aligned to the same genomic region (Chr2D_650894588) in this study. Further analysis annotated a candidate gene TraesCS2D02G600000.2 on Chr2D, which encodes callose synthase and plays an important role in endosperm development. In previous studies, fluorescent dyes were used to investigate the ovaries of rice at different developmental stages. It was found that there is a dynamically changing callose “sheath” outside the endosperm. With the growth and development of the endosperm, it further demonstrates that the dynamic changes of callose play a crucial role in the normal development of the endosperm (Wang et al., 2004). The Chr2D_650894588 Hap.3 was an elite haplotype on seed yield trait. Fei et al. (2022) located the yield regulating SNP of wheat during the late filling stage to the positions of 15.94–18.67 Mb on wheat chromosome 6A, which overlapped with the Chr6A_17541092 locus. These two loci could be identified as the yield-controlling SNPs.

In this study, 15 pleiotropic loci associated with two or more seed related traits were found by genome-wide association analysis. Combined with phenotype, it is obviously that the function of the same locus gene can simultaneously regulate multiple traits, such as increased spike length and seed yield. Therefore, through technological innovation, the full development and utilization of forage resources, and the precise improvement of forage germplasm through molecular design are revolutionary technologies for cultivating super forage varieties.

Conclusion

Seed yield-related traits in P. juncea, a key forage species for grassland restoration and livestock production, are polygenic quantitative traits critical for breeding improvement. GWAS was conducted on a natural population of P. juncea using the wheat reference genome to dissect the genetic basis of eight seed yield traits. We identified 121 significant SNPs, including 19 stable loci detected across two environments, a total of 91 candidate genes were annotated, which involve the synthesis of cell wall polysaccharides and proteins, plant growth and development, photosynthesis, gibberellin regulation, hormone signal transduction, phenylalanine metabolism, and amino acid metabolism processes. The objectives of this study, QTL and the genomic information of seed traits may integrate marker-assisted selection (MAS) in the development of new cultivars.

Supplemental Information

Supplemental Information 1 Supplemental Figures and Tables

We thank the Key Laboratory of Grassland Resources and Key Laboratory of Forage Cultivation, Processing and High Efficient Utilization of the Ministry of Agriculture and Grassland Germplasm Innovation and Sustainable Utilization of Grassland Resources in Inner Mongolia Autonomous Region, Inner Mongolia Agricultural University for completing all experiments in the laboratory.

Additional Information and Declarations

Competing Interests

Author Contributions

Data Availability

The authors declare there are no competing interests.

Zhen Li conceived and designed the experiments, prepared figures and/or tables, and approved the final draft.

Tian Wang performed the experiments, analyzed the data, prepared figures and/or tables, and approved the final draft.

Xiaomin Ren performed the experiments, analyzed the data, prepared figures and/or tables, and approved the final draft.

Feng Han analyzed the data, authored or reviewed drafts of the article, and approved the final draft.

Yingmei Ma analyzed the data, authored or reviewed drafts of the article, and approved the final draft.

Lan Yun conceived and designed the experiments, authored or reviewed drafts of the article, and approved the final draft.

The following information was supplied regarding data availability:

The raw sequencing reads in this study are available in the Sequence Read Archive at NCBI: PRJNA1014568.

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
