# Peer review of "A SNP-based genome-wide association study (GWAS) of seed-yield related traits in Psathyrostachys juncea using wheat as a reference genome"

_PeerJ, doi:10.7717/peerj.19617_

## Round 0.1 · original submission · Major Revisions

Dear Authors

The manuscript cannot be accepted for publication in its current form. It needs a major revision before publication. The authors are invited to revise the paper considering all the suggestions made by the reviewers. Please note that the requested changes are required for publication.

With Thanks

Reviewer 1 ·

Basic reporting

• The paper is generally well-written, but there are minor grammatical and typographical errors that should be corrected to improve clarity.
• The introduction provides sufficient background, but it lacks a strong justification for using wheat as a reference genome. Why was wheat chosen instead of a more closely related species?
• Figures and tables appear relevant, but their clarity and resolution should be checked for readability.

Experimental design

• The research question is well-defined, and the study aims to fill a knowledge gap in P. juncea seed yield genetics.
• The use of 300 accessions across different environments strengthens the study, but more design used for field experiment is not appropriate, author may justify why they used RBD and number of replications need to mentioned.
• The methodology for GWAS is robust, but additional clarification is needed on how population structure (Q and K) was controlled during association analysis.

Validity of the findings

• The statistical analysis appears appropriate, but it is unclear whether any corrections (e.g., Bonferroni or FDR) were applied for multiple testing in GWAS.
• The identification of candidate genes is interesting, but functional validation (e.g., gene expression analysis) is lacking.
• The conclusions are well-supported by the results, but the potential practical applications of these SNPs in P. juncea breeding should be further elaborated.

Additional comments

These are the concern need to be critically addressed with adequate explanation
1. Why was wheat used as a reference genome instead of a more closely related species?
2. How were potential false positives in GWAS addressed? Was a multiple testing correction applied?
3. How stable are the identified SNPs across different environmental conditions? Were genotype-environment interactions considered?
4. What are the potential applications of these SNPs in breeding programs? Have any candidate genes been validated experimentally?
5. Were the accessions used in this study diverse enough to represent the genetic variation in P. juncea populations globally?
6. Line #16-17 While using the term resistant for abiotic stresses care should be taken, and it is better to use tolerant instead resistant.
7. Line #22-23 Authors mentioned that CV for yield trait exceeded 70%, it indicates there are some inherent errors in experimentation. Authors can clearly explain why this much variation.
8. Line # 95 SLAF-seq, For the first time write it completely
9. Line#113 It is not advisable to use Randomized block design for such large number of germplasm accession and also authors have not mentioned row length, number of replications???? Instead RBD authors would have used alpha lattice design
10. What was the minor allele frequency set for the GWAS, ???

Annotated reviews are not available for download in order to protect the identity of reviewers who chose to remain anonymous.

Reviewer 2 ·

Basic reporting

I have read manuscript #112251, entitled "A SNP-based genome-wide association study (GWAS) of seed yield related traits in Psathyrostachys juncea using wheat as a reference genome". The manuscript presents genetic signals associated with seed yield
in P. juncea germplasm through a genome-wide association study (GWAS). Psathyrostachys juncea is a perennial grass used as forage in norther of China of cold uplands and semiarid deserts. Seed yield is an important traits, and many attempts have made to improve seed biomass. Until now, recurrent selection procedures have been widely employed in the genetic improvement of populations and development of cultivars in Psathyrostachys juncea. However, few studies in genetic have focus on this grasses, especially in complex of seed yield. The manuscript fill its gap, and presents some new findings and has novelty. Although the topic is interesting, several issues prevent recommendations for acceptance in the present form. some changes are needed to have a high-quality manuscript.

Experimental design

no comment. but need add more information of field managements.

Validity of the findings

no comment

Additional comments

1. “Abstract” section. Line 16 “ It is also a valuable germplasm for improving cold and drought resistant of forage grass species”, should delete. I think Psathyrostachys juncea as a perennial grass for forage is enough, and very few used as parent for improve resistant for other grasses.
2. “Abstract” section methods, line 20 “Two clonal populations established from”, should delete.
3. “Abstract” section results, line 35 “and wheat crop”, should delete. I suggest the finding of this study provide information of genetic basis associated with seed traits in Psathyrostachys juncea and could contribute to marker-assisted selection for breeding new type cultivars of P. juncea, but not for wheat crop.
4. “Materials and methods” section. Line 115. “ field management was consistent with local field production management”. I suggest the field management need more details. For example, in 2022, ** kg ha-1 17-17-17 (N-P2O5-K2O) fertilizer was applied before transplanting. On August, ** kg ha-1 urea (46-0-0, N-P2O5-K2O) was applied to promote rapid establishment. Supplementary irrigation was given to avoid any drought stress during the establishment phase. et al. Because fertilization and irrigation (time and frequence) have a significant impact on seed development. So I suggest there need more description of plot management. And also how deal with weeds.
5. “Materials and methods” section. Line 157. “bread wheat was selected as the reference genome for SNP localization”. In there, author should be made clear why the wheat genome is used.
6. “conclusion” section. I suggest the conclusion should be rewritten. First sentence, introduction the species and seed traits. Second sentence, population of p. juncea and the method of. Third or/and four sentence, the core results. Last, the objectives of this study, QTL and the genomic information of seed traits may integrate marker-assisted selection (MAS) in the development of new cultivars.

Reviewer 3 ·

Basic reporting

Dear Editor,

I have completed the review of the manuscript titled "A SNP-based genome-wide association study (GWAS) of seed yield-related traits in Psathyrostachys juncea using wheat as a reference genome." The study provides valuable insights into the genetic basis of seed yield in P. juncea, identifying significant SNPs and candidate genes that could contribute to future breeding programs.

The research is well-structured, methodologically sound, and contributes to the field of plant genetics and molecular breeding. However, I noticed a few minor English writing errors throughout the text. I recommend a brief language revision to ensure clarity and readability.

Given its scientific merit and relevance, I am pleased to recommend the manuscript for acceptance in PeerJ.

Best regards,

Experimental design

This study were employed a genome-wide association study (GWAS) to identify genetic signals associated with seed yield-related traits in Psathyrostachys juncea.

A total of 300 accessions from different geographic regions were evaluated under two environmental conditions over two consecutive years.

8 seed yield-related traits were measured, and genotyping was performed using SLAF-seq technology, generating 84,024 high-quality SNP markers.

Population structure and linkage disequilibrium were analyzed, followed by GWAS using a mixed linear model (MLM) to control for population structure and kinship.

Significant SNP-trait associations were identified, and candidate genes were annotated based on wheat genome references.

The findings provide valuable insights into the genetic basis of seed yield in P. juncea, supporting future molecular breeding efforts.

Validity of the findings

The study's findings are supported by a robust experimental design, including a diverse germplasm set, multi-environment trials, and high-throughput SNP genotyping.

---

## Round 0.2 · Major Revisions

Dear Authors
The manuscript cannot be accepted for publication in its current form. It needs a major revision before publication. The authors are invited to revise the paper, considering all the suggestions made by the reviewers. Please note that the requested changes are required for publication.
With Thanks

Reviewer 1 ·

Basic reporting

Clear and unambiguous
In revision, all the suggestions were incorporated by the authors

Experimental design

In the revision, all the suggestions were incorporated by the authors

Validity of the findings

In the revision, all the suggestions were incorporated by the authors

Additional comments

In the revision, all the suggestions were incorporated by the authors., Therefore, I don't have further comments

·

Basic reporting

no comment

Experimental design

no comment

Validity of the findings

no comment

Additional comments

This article focuses on the GWAS of seed yield related traits of Psathyrostachys juncea based on SNP, with clear overall logic and detailed content.
1. The paper is generally well-written, but there are minor grammatical and typographical errors that should be corrected to improve clarity.
2. Figures and tables appear relevant, but their clarity and resolution should be checked for readability.
3. “Introduction” section. Line 39 “Psathyrostachys is originated in Eurasia...”, it is recommended to modify it to “Psathyrostachys originated in Eurasia...”.
4. “Introduction” section. Line 46-47 “After been introduced into North America P. juncea also known as Russian wildrye” has a syntax error. It is recommended to modify it to “After being introduced into North America, P. juncea is also known as Russian wildrye.”
5. “Results” section. Line 201-203 “the coefficient of variation of SY was the largest, with an average of 72.41%; and the coefficient of variation of RTL was the smallest, at 14.27%.” It is recommended to modify it to "Coefficient of variation (CV) for seed yield (SY) was highest (72.41%), while RTL showed the lowest CV (14.27%).”

Reviewer 5 ·

Basic reporting

Clear and unambiguous, professional English used throughout.
Yes, the manuscript is written in clear, professional English with minimal grammatical errors. Technical terms are well-defined, and the language is accessible for an international audience.

Literature references, sufficient field background/context provided.
The introduction provides a solid background on P. juncea and GWAS applications in forage grasses. However, some recent studies on wheat GWAS could be cited for comparative context.

Professional article structure, figures, tables.
The structure follows standard scientific reporting, but figures (e.g., Manhattan plots) could be simplified for clarity. Tables are well-organized but should include units for all measurements.

Raw data shared.
Yes, raw sequencing data is deposited in NCBI (PRJNA1014568). However, phenotypic data availability should be explicitly confirmed.

Self-contained with relevant results to hypotheses.
The results support the hypotheses, identifying SNPs and candidate genes linked to seed yield traits. However, the discussion could better connect findings to broader biological mechanisms.

Experimental design

Original primary research within Aims and Scope of the journal.
Yes, the study presents original GWAS research on P. juncea, aligning with journals focused on plant genetics, genomics, and breeding.

Research question well defined, relevant & meaningful. It is stated how research fills an identified knowledge gap.
The research question is clearly defined—identifying genetic markers for seed yield traits in P. juncea. It fills a gap in forage grass genomics, where such studies are scarce compared to crops like wheat.

Rigorous investigation performed to a high technical & ethical standard.
The study employs robust methods (SLAF-seq, multi-environment trials, GWAS) and adheres to ethical standards, though replication details could be clarified.

Methods described with sufficient detail & information to replicate.
Methods are generally well-documented, but additional specifics on field trial replication, SNP filtering parameters, and GWAS covariates would improve reproducibility.

Validity of the findings

Impact and novelty not assessed. Meaningful replication encouraged where rationale & benefit to literature is clearly stated.
The study's novelty lies in applying wheat-referenced GWAS to P. juncea, advancing forage grass genetics. While impact isn't explicitly assessed, the identified SNPs and candidate genes provide a foundation for molecular breeding. Replication in independent populations would strengthen validity. The rationale for using wheat as a reference genome should be expanded to better highlight its benefits for understudied species.

All underlying data have been provided; they are robust, statistically sound, & controlled.
Sequencing data is publicly available (NCBI PRJNA1014568), and phenotypic analyses appear statistically rigorous (BLUP, ANOVA, FDR correction). However, raw phenotypic data should be explicitly shared, and GWAS covariates (e.g., kinship, population structure) should be detailed to confirm robustness.

Conclusions are well stated, linked to original research question & limited to supporting results.
Conclusions directly address the research question, summarizing key SNPs and candidate genes for seed yield traits. They avoid overinterpretation, though a brief discussion on translational potential (e.g., marker-assisted breeding) would better connect findings to practical applications.

Additional comments

This study presents a comprehensive GWAS of seed yield-related traits in Psathyrostachys juncea, leveraging wheat as a reference genome. The research identifies significant SNPs and candidate genes, offering valuable insights for molecular breeding. The experimental design is robust, with multi-environment phenotyping and high-quality SNP data. However, the manuscript would benefit from clearer methodological descriptions and deeper discussion of the biological relevance of candidate genes. Overall, the study is well-conducted but requires refinements in presentation and interpretation.
1. Clarify the rationale for selecting wheat as the reference genome, given the phylogenetic distance between wheat and P. juncea, and discuss potential limitations.
2. Provide more details on the SLAF-seq protocol, including sequencing depth, coverage, and quality control metrics, to ensure reproducibility.
3. The manuscript mentions 300 accessions but does not detail how genetic diversity was preserved or whether any subpopulations were underrepresented.
4. The coefficient of variation for seed yield per plant (SY) is notably high (70%). Discuss potential causes, such as environmental heterogeneity or genetic instability.
5. The correlation analysis (Figure 2) should include a discussion on whether the observed trait correlations are consistent with prior studies in related species.
6. The population structure analysis (Figure 3) groups accessions by geographical origin. Justify whether this structure was accounted for in GWAS to avoid spurious associations.
7. The LD decay analysis (Figure 3c) should include a comparison with other forage grasses to contextualise the findings.
8. The Manhattan plots (Figure 4) are visually cluttered. Consider simplifying or separating the traits into individual plots for clarity.
9. Table 1 lacks units for some traits (e.g., RTL, SL). Ensure all measurements are explicitly defined.
10. The ANOVA (Table 2) shows high heritability for traits like TKW. Discuss whether this is typical for forage grasses or if it suggests strong genetic control.
11. The SNP distribution (Table 3) shows uneven coverage across subgenomes. Address whether this bias could affect GWAS results.
12. The candidate gene annotations (Table 4) are descriptive but lack functional validation. Suggest future work, such as qPCR or CRISPR, to confirm gene roles.
13. The haplotype analysis (Figures 6–9) is insightful but should include statistical significance tests for phenotypic differences between haplotypes.
14. The discussion of pleiotropic SNPs (e.g., Chr7A_675095747) should explore whether these represent true pleiotropy or linked genes.
15. The manuscript cites prior QTL studies but does not systematically compare their physical positions with the GWAS hits. A comparative table would strengthen the discussion.
16. The candidate gene TraesCS2D02G600000.2 (callose synthase) is linked to SY. Discuss its known role in seed development in other species.
17. The F-box protein (TraesCS7A02G483300.1) is associated with multiple traits. Expand on its potential regulatory mechanisms in yield-related pathways.
18. The RNA polymerase II gene (TraesCS5D02G308200.1) is annotated but its relevance to seed yield is unclear. Provide a hypothesis for its role.
19. The discussion of environmental effects (e.g., Baotou vs. Hohhot) is superficial. Include soil or climate data to support claims about phenotypic variation.
20. The manuscript notes high genetic diversity but does not discuss how this impacts GWAS power or resolution.
21. The GWAS thresholds (P < 1.27 × 10^−6 and 10^−7) are stringent. Justify these cut-offs and discuss whether any suggestive SNPs were excluded.
22. The pleiotropy analysis should address whether the observed multi-trait SNPs are driven by shared genetic pathways or sampling bias.
23. The candidate gene list includes many uncharacterised proteins. Prioritise discussing those with known functional domains or homologs in model species.
24. The raw data availability statement is clear, but the manuscript should confirm whether all phenotypic data are also deposited.
25. The introduction cites P. juncea's economic importance but does not quantify its global or regional use. Provide statistics to underscore its relevance.
26. The methods section omits details on field trial replication. Specify the number of replicates per accession and environment.
27. The BLUP analysis is mentioned but not described in sufficient detail. Clarify how genotype-by-environment interactions were modelled.
28. The manuscript refers to "favorable alleles" but does not define this term. Specify whether these are based on phenotypic means or breeding values.
29. The discussion of OsCSLD4 homologs in P. juncea is intriguing but lacks experimental evidence. Suggest complementation tests in future work.
30. The candidate gene TraesCS6A02G035700.1 (OFP14) is linked to SL and SY. Discuss its role in transcriptional regulation and yield traits in other crops.

·

Basic reporting

The manuscript meets basic reporting requirements of this journal. I made a few editorial suggestions in the attached marked-up manuscript. I suggest to use hyphen for the compound adjective "seed-yield" in the manuscript title.

Experimental design

The experimental design was well suited to address important questions about genetic variation for seed-yield traits in Psathyrostachys. The only major problem in the experimental design was in the comparison of results obtained using DNA sequence alignments to the wheat A, B and D subgenomes. It simply does not make sense to compare Psathyrostachys diversity statistics between the wheat subgenomes. This should be removed from the manuscript.

Validity of the findings

I believe that the overall validity of the findings is good, except that it did not make any sense to compare Psathyrostachys diversity statistics between the wheat subgenomes.

---

## Round 0.3 · accepted · Accept

Dear Authors,

I am pleased to inform you that the manuscript has improved after the last revision and can be accepted for publication.

Congratulations on accepting your manuscript, and thank you for your interest in submitting your work to PeerJ.

With Thanks


Examples:

Two minor corrections to the abstract as follows: Line 15: "an popular" should read "a popular"; Line 28 the abbreviation "K" should be defined when first used.

·

Basic reporting

no comment

Experimental design

no comment

Validity of the findings

no comment

Additional comments

After carefully reading, I think that the manuscript has been improved a lot.

·

Basic reporting

The general use of English is technical and professional throughout, with adequate references.

Experimental design

The experimental design is appropriate.

Validity of the findings

The results are scientifically sound and meaningful.